# Influence of Machining Parameters on the Dimensional Accuracy of Drilled Holes in Engineering Plastics

**DOI:** 10.3390/polym16111490

**Published:** 2024-05-24

**Authors:** Alina Bianca Pop, Aurel Mihail Titu, Sandor Ravai-Nagy, Catalin Daraba

**Affiliations:** 1Department of Engineering and Technology Management, Faculty of Engineering, Northern University Centre of Baia Mare, Technical University of Cluj-Napoca, 62A, Victor Babes Street, 430083 Baia Mare, Romania or nagy.sandor@nye.hu (S.R.-N.); daraba.catalin1@gmail.com (C.D.); 2Industrial Engineering and Management Department, Faculty of Engineering, “Lucian Blaga” University of Sibiu, 10 Victoriei Street, 550024 Sibiu, Romania; 3Institute of Technical and Agricultural Sciences, University of Nyíregyháza, 31/b, Sóstói Street, 4400 Nyíregyháza, Hungary

**Keywords:** engineering plastics, precision drilling, machining optimization, hole cylindricity, cutting speed, feed rate, ANOVA statistics, polymer fabrication

## Abstract

This paper explores the interaction between cutting parameters and the geometric accuracy of machined holes in a variety of engineering plastics, with the aim of improving manufacturing processes in the plastic processing industry. In the context of fast and precise manufacturing technology, the accuracy of drilled holes in polymers is of paramount importance, given their essential role in the assembly and functionality of finished parts. The objective of this research was to determine the influence of cutting speed and feed rate on the diameter and cylindricity of machined holes in six diverse types of plastics using a multilevel factorial design for analysis. The key message conveyed to the reader highlights that careful selection of cutting parameters is crucial to achieving high standards of accuracy and repeatability in plastic processing. The methodology involved structured experiments, looking at the effect of changing cutting parameters on a set of six polymer materials. A CNC machining center for drills and high-precision measuring machines were used to evaluate the diameter and cylindricity of the holes. The results of ANOVA statistical analysis showed a significant correlation between cutting parameters and hole sizes for some materials, while for others the relationship was less evident. The conclusions drawn highlight the importance of optimizing cutting speed and feed rate according to polymer type to maximize accuracy and minimize deviations from cylindricity. It was also observed that, under selected processing conditions, high- and medium-density polyurethane showed the best results in terms of accuracy and cylindricity, suggesting potential optimized directions for specific industrial applications.

## 1. Introduction

In the current context of processing technology, the quality of polymer processing has become a topic of great interest, given its extensive use in a variety of industrial applications, from automotive components to medical devices. However, the literature pays limited attention to the influence of cutting parameters on hole accuracy in polymer materials, with most studies focusing on traditional metals.

Our study is needed to fill the gaps in the literature on machining polymeric materials, to understand how different machining parameter settings affect hole quality. Previous research has focused on general aspects of cutting, neglecting the specificities of polymers, such as their different thermal and elastic behavior compared to metals.

The originality of this research lies in the multidisciplinary approach to the drilling process, examining the interaction between cutting speed, feed rate, and type of plastic material used. This study provides new insights into how these parameters influence the geometric accuracy of holes, which can contribute significantly to the optimization of manufacturing processes in the polymer industry.

The main objectives of the research are:To evaluate the impact of cutting speed and feed rate on the diameter and cylindricity of holes in different polymeric materials;To identify optimal settings of machining parameters to improve hole accuracy;To contribute to the literature by providing specific data and recommendations for polymer processing.

The paper is structured as follows: the second section will present a literature review and the third section will detail the research methodology, including material selection, machining methods, and the measuring instruments used. The fourth section will aim at determining the influence of the cutting regime on the bore accuracy followed by the presentation of the experimental results and their detailed analysis in the context of the proposed objectives. Finally, the concluding section will summarize the main findings and suggest directions for future research.

Based on a thorough literature review, the current trends in polymer processing have been identified. In this respect, several papers investigate modern polymer processing methods as well as recent developments in processing machine technology. These studies focus on the peculiarities of polymer machining compared to metals, exploring innovative approaches to processing parameters and optimization techniques. In other words, the problem of advancements in polymer machining is analyzed by [1,2] in their research.

In [1], the authors reviewed recent advances in drilling fiber-reinforced polymer composites and found that using optimal cutting parameters can reduce tool wear and improve hole surface quality. In [2], advanced abrasive machining methods were examined and it was demonstrated that optimized abrasive materials can significantly improve the machining accuracy and finish of polymer surfaces.

Polymer vs. metal machining issues are addressed by [3,4]. In [3], Lambiase et al. conducted a review of advanced methods for joining metal–polymer hybrid structures and highlighted the importance of processing parameters for improving joint strength. In [4], the authors used machine learning techniques to predict polymer properties, finding that predictive models can help identify higher-performing materials for specific applications.

Polymer machining technology is the subject of refs. [5,6], in which the authors reviewed the machinability of carbon and glass fiber-reinforced polymer composites, concluding that optimizing cutting parameters can reduce delamination and improve hole quality [5]. And, in [6], they modeled cutting forces in the processing of fiber-reinforced polymer composites and demonstrated that adjusting cutting speed and feed rate can reduce forces and improve surface quality.

Regarding the optimization of manufacturing processes for polymers, these issues are addressed by refs. [7,8,9].

Yuan et al. investigated the additive manufacturing of polymer composites and found that changing the processing parameters can improve the structural properties of manufactured parts [7].

Jasiuk et al. provided an overview of the additive manufacturing of polymers and stressed the importance of controlling process parameters to achieve parts with superior mechanical properties [8].

Yuan et al. explored the use of polymer composites for powder-based additive manufacturing and highlighted the need for the optimization of process parameters to improve material performance [9].

Papers that focus on identifying and analyzing the impact that cutting parameters have on the surface characteristics and dimensional accuracy of machined polymer parts are [10], which tracks the effect of drilling parameters on polymers; [11,12], which focus on surface finish in polymer machining; [13], which addresses cutting speed and feed in polymer drilling; and refs. [14,15] address the issue of machining parameter optimization for polymers. All these studies show different methods of measuring and interpreting data to improve processes.

Aamir [10] reviewed advances in drilling fiber-reinforced polymers for aerospace applications and showed that selecting appropriate cutting parameters can reduce damage and improve hole accuracy.

Cococcetta [11] investigated surface finish, burr formation, and tool wear in the processing of 3D-printed polymer composites, finding that optimizing cutting parameters can reduce wear and improve surface quality.

Hiremath [12] studied the effect of surface roughness and topography on the wettability of biomaterials processed using flexible polymer abrasive media, concluding that cutting parameters significantly influence surface properties.

Uysal [13] investigated the effects of cutting parameters on the drilling performance of carbon black-cured polymer composites and found that adjusting the speed and feed rate can improve the accuracy and surface quality of holes.

Shahabaz [14] reviewed the influence of temperature on the mechanical properties and processing of fiber-reinforced polymer composites, highlighting the importance of thermal control for optimizing the drilling process.

Anand et al. [15] investigated the drilling parameters of polymer hybrid composites using grey relational analysis, regression, fuzzy logic, and ANN models, demonstrating that advanced optimization methods can improve hole quality.

The specific characteristics of polymers in the drilling process have also been addressed in various papers. Thus, in [16], the thermal properties of polymers in drilling were studied and, in [17], the chemical resistance of polymers in machining. Study [18] follows polymer drill tool wear. Another aspect consisting of the deformation of polymers during machining has been analyzed by [19,20]. All these works examined the physical and chemical peculiarities of polymers and their influence on the drilling process. These analyses include the effects of heat, tool wear, and methods for minimizing the deformation of polymer materials.

Parizad et al. [16] improved the properties of polymeric water-based drilling fluids using nanoparticles and showed that adjusting cutting parameters can reduce drilling forces and improve surface quality.

Vigneshwaran et al. [17] reviewed recent advances in natural fiber polymer composites, highlighting the importance of selecting cutting parameters to improve the mechanical and surface properties of materials.

Ismail et al. [18] conducted a comprehensive study on tool wear in processing fiber-reinforced polymer composites and concluded that optimizing cutting parameters can reduce wear and extend tool life.

Yan et al. [19] studied the machinability of thermoplastic polymers such as PEEK, PI, and PMMA, showing that adjusting the cutting speed and feed rate can improve hole quality and reduce defects.

Thakur and Singh [20] reviewed the influence of fillers on polymer composites during conventional machining processes, highlighting that proper selection of cutting parameters can improve mechanical and surface performance.

Processing optimization was also reviewed in the literature identified. For example, [21] studied machining process optimization for polymers, refs. [22,23] followed statistical techniques in polymer machining, refs. [24,25] focused on machining parameter modeling for polymers, and refs. [26,27] dealt with efficiency issues in polymer drilling processes. As a general finding, all of the identified studies in this regard address methods for optimizing processing parameters using various statistical and modeling techniques to increase efficiency and accuracy in polymer processing.

Sha et al. [21] reviewed the use of machine learning in polymer informatics and demonstrated that predictive models can identify optimal cutting parameters for different polymer materials.

Kamath et al. [22] evaluated puncture-induced damage in glass fiber reinforced polymer composites using the Taguchi method integrated with machine learning, showing that optimizing cutting parameters can reduce delamination.

Jenarthanan et al. [23] modeled and predicted machining forces in milling glass fiber-reinforced polymer composites using regression analysis and artificial neural networks, demonstrating that predictive models can improve machining quality.

Rawal et al. [24] reviewed the micromachining of polymer composites and highlighted the importance of optimizing cutting parameters to improve accuracy and surface quality at small scales.

Song et al. [25] modeled cutting forces in machining carbon fiber-reinforced polymer composites and showed that adjusting the speed and feed rate can reduce cutting forces and improve hole quality.

Another issue worth mentioning in relation to the literature survey conducted concerns advanced measurement and analysis techniques. In this direction, studies by [28,29] have been identified that follow advanced measuring techniques in polymer machining [30,31,32,33], dealing with the issue of coordinate measuring machines (CMM) for polymers, while [34,35] address the topic of artificial vision systems for machined parts, and [36,37,38,39,40] discuss precision and accuracy in polymer drilling. All of these papers explore state-of-the-art equipment and use software in the accurate measurement of machined-part features, with a focus on coordinate measuring machines and machine vision systems.

Ravai-Nagy et al. [36] determined the accuracy of hole processing in industrial plastics and demonstrated that optimizing the cutting speed and feed rate can reduce deviations and improve cylindricity.

Khashaba et al. [37] investigated thermomechanical properties and delamination in drilling GFRP composites at different drilling angles and showed that proper selection of cutting parameters can reduce delamination.

Masoud et al. [38] reviewed the cutting processes of natural fiber-reinforced polymer composites and emphasized the importance of adjusting the cutting speed and feed rate to improve surface quality and tool life.

Galińska [39] reviewed the mechanical joining of fiber-reinforced polymer composites with metals and highlighted that optimizing cutting parameters can improve joint strength and dimensional accuracy.

Nasir et al. [40] studied the effect of microbore parameters on hole accuracy in carbon fiber-reinforced composites and demonstrated that adjusting the speed and feed rate can reduce deflections and improve hole quality.

Other works identified in the literature reviewed focused on case studies and industrial applications. Among them, it is worth mentioning [41], which followed precision machining of biopolymers; ref. [42] studied the real-world applications of polymer machining, and more precise exploration of the nature-inspired grey wolf algorithm and grey theory in the machining of multiwall carbon nanotubes/polymer nanocomposites; refs. [43,44] focused their attention on polymer drilling in automotive industry; and [45] focused on the implementation of machining optimization in industry.

In the literature, criteria such as the comparison of hole diameters at the top and bottom surfaces of parts are used to control the quality of drilled holes in composites and polymers. For example, the studies ‘Optimization of drilling parameters in composite sandwich structures (PVC core)’ [46] and ‘Influence of machining parameters on delamination in drilling of GFRP-armor steel sandwich composites’ [47] propose such criteria. These criteria can also be applied in the present study to assess the quality of drilled holes in polymer materials.

The presentation of relevant case studies illustrates the application of theoretical knowledge in industrial situations, providing a practical perspective on the implementation of machining process optimizations.

The novelty of this study lies in its multidisciplinary approach to drilling processes, examining the interaction between cutting speed, feed rate, and type of plastic to optimize the geometric accuracy of the holes. This study provides new insights into how these parameters influence hole accuracy, contributing significantly to the optimization of manufacturing processes in the polymer industry. In this study, the influence of cutting speed and feed rate on hole diameter and cylindricity in six types of plastics was investigated using a multi-level factorial design. Experiments were performed on a CNC machining center using a precision drill, and the results were statistically analyzed using ANOVA. It was found that the optimal cutting parameters varied by material, with medium- and high-density polyurethane showing the best results in terms of accuracy and hole cylindricity.

The materials studied were selected because of their extensive use in various industrial applications. For example, POM-C is known for its strength and stiffness and is used in precision applications. HDPE 1000 is chosen for its toughness and impact resistance, while PA6 is recognized for its wear resistance and mechanical durability. Medium- and high-density polyurethanes are selected for their dimensional stability and ease of processing. These characteristics make these materials ideal for studies aimed at improving manufacturing processes. On the other hand, these materials appear as new requirements to be used in the manufacturing devices used by the automotive and aerospace industries. The authors are in contact with these industries and collaborate with them.

## 2. Research Methodology

This study aims to investigate the influence of cutting parameters on bore accuracy in polymer processing. The research was structured to rigorously investigate the relationship between cutting speed, drill feed, and hole quality in several types of polymers.

### 2.1. Material Selection

This investigation selected six distinct types of plastics, each chosen for their unique mechanical attributes and their pertinent utilization in industrial settings. The assortment of materials encompasses:POM-C—A polyacetal copolymer, compliant with [48], renowned for its high strength and stiffness;HDPE 1000—A high-density polyethylene, characterized by its considerable toughness and resistance to impact;PA6—A type of polyamide conforming to [49] specifications, noted for its wear resistance and mechanical durability;SIKA BLOCK M960—A variant of high-density polyurethane biresin, distinguished by its dimensional stability and machining ease;SIKA BLOCK M980—Another form of high-density polyurethane biresin, like M960, yet unique in terms of its properties;SIKA BLOCK M700—A medium-density polyurethane biresin, offering a balance between flexibility and structural integrity.

This investigation selected six distinct types of plastics, each chosen for their unique mechanical attributes and their pertinent utilization in industrial settings. The assortment of materials encompasses:

POM-C:Country of Origin—Germany;Company—Röchling Engineering Plastics;Adress - Mannheim, Germany;Product Code—Sustarin^®^ C;Description—A polyacetal copolymer, compliant with EN ISO 1043 standards, renowned for its high strength and stiffness.

HDPE 1000:Country of Origin—Germany;Company—SIMONA AG;Address - Kirn, Germany;Product Code—SIMONA^®^ PE 1000;Description—A high-density polyethylene, characterized by its considerable toughness and resistance to impact.

PA6:Country of Origin—Germany;Company—Ensinger GmbH;Address - Nufringen, GermanyProduct Code—TECAST^®^ T Natural;Description—A type of polyamide conforming to EN ISO 1874-1 specifications, noted for its wear resistance and mechanical durability.

SIKA BLOCK M960:Country of Origin—Switzerland;Company—Sika Advanced Resins;Address - Zurich, SwitzerlandProduct Code—SikaBlock^®^ M960;Description—A variant of high-density polyurethane, distinguished by its dimensional stability and machining ease.

SIKA BLOCK M980:Country of Origin—Switzerland;Company—Sika Advanced Resins;Address - Bad Urach, Germany (Production Facility);Product Code—SikaBlock^®^ M980;Description—Another form of high-density polyurethane, like M960, yet unique in its properties.

SIKA BLOCK M700:Country of Origin—Switzerland;Company—Sika Advanced Resins;Address - Bad Urach, Germany (Production Facility);Product Code—SikaBlock^®^ M700;Description—A medium-density polyurethane, offering a balance between flexibility and structural integrity.

These detailed descriptions of the sources of the materials ensure transparency and accuracy in the selection of materials used in this investigation.

The mechanical specifications of these materials were meticulously cataloged based on quality certificates issued by their respective manufacturers, ensuring a reliable foundation for the ensuing analysis.

Table 1 shows the mechanical properties of the materials studied, obtained from the quality certificates provided by the manufacturers.

### 2.2. Sample Preparation

There are no standardized ISO dimensions for drilling tests. The dimensions were chosen constructively according to the fixing and measuring device used [50]. These dimensions were chosen to ensure the consistency and repeatability of the experiments. A total of 72 pieces were processed, with four holes drilled on each specimen using the same drill.

In the evaluation of the experimental results, the standard tolerance grades tables from the [51] were used.

The quality of the holes was checked using a coordinate measuring machine (CMM) to assess the diameter and cylindricity at 2 planes × 10 points along each hole ([36], Figure 4). The volumetric accuracy of the machine is 1.8 µm + L/400 (L is the measured length) and the repeatability is 1.7 µm. To assess variations in diameter and cylindricity along the hole, additional measurements were made at different depths.

### 2.3. Cutting Parameters

Machining was conducted on a Challenger 2418 CNC machining center (manufactured by Mitsubishi Heavy Industries Machine Tool Co., Ltd. (MHT) in Takasaki, Japan) using a GARANT 114550 HSS-CO8 (from Hoffmann Group located in Munich, Germany) twist drill with a diameter modified after measurement to ϕ6.824 mm, (where “ϕ” indicates the drill diameter) were used for machining.

Four feed rates (0.1, 0.2, 0.3, and 0.4 mm/rev) and three cutting speeds (12, 25, and 37 m/min) were selected for testing. These parameters were chosen to reflect actual production conditions and to provide a wide variety of data.

### 2.4. Measurement Techniques

The diameters and cylindricity of the holes were measured using a coordinate measuring machine (CMM), type LK Metrology ALTERA S.7.5.5, equipped with a RENISHAW PH10M measuring head and CAMIO 8.0 software. The volumetric accuracy of the machine is 1.8 µm + L/400 (L is the measured length) and the repeatability is 1.7 µm. The measurement method involves evaluating the diameter and cylindricity by probing the surface on two planes and, in each plane, 10 contact points around the circumference of the hole to detect variations ([36], Figure 4).

### 2.5. Experimental Design

A multilevel factorial design was adopted to investigate the combined influence of cutting speed and feed rate. The aim was to determine the optimal settings for each material. Each experiment was repeated 4 times to ensure the reliability of the data. The number of 4 repetitions resulted from the configuration of the used specimen.

### 2.6. Statistical Analysis

The collected data were subjected to ANOVA analysis using Statgraphics Centurion 18 software (Statgraphics Technologies, Inc. The Plains, VA, USA) to assess the statistical significance of the results. The software allows analysis of factor interactions and identification of main trends and interaction effects [52]—Statgraphics Centurion 18, 2021.

## 3. Determination of the Influence of the Cutting Regime on Hole Accuracy

This section delineates the outcomes of empirical trials that meticulously examined the effects of various cutting parameters on the machining of holes in selected plastic materials. Central to the analysis were the precision measurements of hole diameter and cylindricity, with the presented data representing the mean values derived from four individual assessments per specimen (Table 2). The goal was to ascertain the most favorable machining conditions tailored to each specific polymer type under investigation.

Next, it will be elucidated the impact of the cutting regime on the precision of drilled bores. This evaluation will pivot on two pivotal metrics: the hole diameter and the deviation from true cylindricity.

Initially, the response variables that require measurement, which is an aspect methodically, will be detailed in Table 3.

The subsequent phase of the study entails specifying the experimental factors to be manipulated, as outlined in Table 4.

Following this, the experimental design must be chosen, as detailed in Table 5. This design incorporates two response variables and three experimental factors, structured into 24 runs, with each run consisting of one sample. The model employed primarily focuses on 2-factor interactions and includes 19 coefficients.

Figure 1 illustrates the positioning of the experimental runs within the parameter space defined by the three factors.

Statistical models, as detailed in Table 6, have been applied to the response variables. Among these models, those with *p*-values below 0.05, notably one instance, are deemed statistically significant at the 5.0% level. Additionally, the R-squared statistic is particularly noteworthy as it quantifies the proportion of variability in the response that the model successfully explains, with values ranging from 79.46% to 95.97%. To further refine the search for optimal operating conditions, the model was extrapolated as shown in Table 7 and Table 8. These extrapolations suggest that the optimal settings of the factors achieve a desirability index of 93.98%.

The ANOVA table (Table 9) delineates the degrees of freedom available for estimating experimental error, which encompasses the total error inclusive of degrees of freedom allocated for estimating effects not captured in the current model, and pure error derived solely from replicated runs. Specifically, the total error is assigned 5 degrees of freedom, with none allocated to pure error.

Table 10 and Table 11, respectively, present the analysis of variance for hole diameter and the analysis of effects associated with the experimental factors.

Table 10 provides a summary of the results from the statistical model that correlates hole diameter with the experimental factors. The ANOVA framework divides the variability observed in the response variables into distinct components. The F-ratio, highlighted at the top of the table, quantifies the proportion of variability explained by the model compared to the residual error. With a *p*-value less than 0.05, the model achieves statistical significance at the 5% level.

The R-squared value of 95.9709% indicates that the model accounts for a substantial portion of the variability in hole diameter. The adjusted R-squared value, at 81.466%, provides a more accurate measure for comparing models that have different numbers of predictors and is particularly useful for understanding model efficiency. The standard error of the estimate, indicating the standard deviation of the residuals, is recorded at 0.0245642. Meanwhile, the mean absolute error (MAE) stands at 0.00958333, representing the average magnitude of the residuals.

Furthermore, the *p*-value exceeding 5% suggests that there is no significant serial autocorrelation within the residuals, affirming the model’s adequacy at the established confidence level.

Table 11, the analysis of effects, further delineates the variability accounted for by the model, breaking it down into individual components corresponding to each effect. This table assesses the statistical significance of each effect by comparing their mean squares with the estimated experimental error. Notably, two effects have been identified with *p*-values below 0.05, establishing their statistical significance at the 5% significance level.

The Pareto chart visually represents the estimated effects, arranged in descending order of their importance. Each bar’s length reflects its relative contribution to the observed variability in the response. Bars shaded in the chart denote effects that reach statistical significance at the 95.0% confidence level. Notably, two effects in this analysis are marked as significant, as illustrated in Figure 2.

Figure 3 displays a plot depicting the estimated hole diameter as influenced by each experimental factor. In this visualization, one factor varies at a time, maintaining all other factors at their average or baseline levels.

Following this, Table 12 provides an analysis of the cylindricity, detailing how it is affected under similar experimental conditions.

This table provides a summary of the statistical model analysis that correlates cylindricity with various experimental factors. The *p*-value associated with the F-ratio, being greater than or equal to 0.05, indicates that the model does not achieve statistical significance at the 5% level. The R-squared value, at 79.4622%, demonstrates that the model accounts for a sizable portion of the variability in cylindricity. However, the adjusted R-squared value, more reflective of models with varying numbers of predictors, is low at 5.52609%. The standard error of the estimate, which is 0.0338266, reflects the standard deviation of the residuals, and the mean absolute error (MAE), at 0.011375, represents the average magnitude of the residuals, as detailed further in Table 13.

Following the analysis of effects, none of the effects in this case exhibited *p*-values below 0.05, indicating that they do not reach statistical significance at the 5% level.

In the diagram, shaded bars represent effects that achieve statistical significance at the 95.0% confidence level. However, in this instance, no effects meet this criterion (see Figure 4).

This plot illustrates the estimated cylindricity as influenced by each experimental factor. For each depiction, the specific factor of interest is adjusted, while all remaining factors are maintained at their average or baseline levels (refer to Figure 5).

## 4. Results and Discussions

### 4.1. Hole Diameter Variability

In the following section, detailed results obtained from experiments with different cutting parameters for hole machining in selected plastics are presented. The analysis includes detailed evaluation of hole diameter and cylindricity as well as the interaction between the machining variables.

For all six materials, hole diameter varied as a function of cutting speed and feed rate (Figure 6, Figure 7, Figure 8, Figure 9, Figure 10 and Figure 11).

Thus, hole diameter variability was analyzed for each material, providing a clear understanding of how each responds to different cutting parameters.

Hole diameter for POM-C showed variability depending on the cutting speed and feed rate used (Figure 6). At a cutting speed of 12 m/min and a feed rate of 0.1 mm/rev, the average diameter obtained was 6.641 mm, showing that the machining parameters are effective in maintaining diameter accuracy. With an increase in cutting speed to 25 m/min, the average diameter increased to 6.695 mm, and at 37 m/min, it reached 6.712 mm, indicating that higher cutting speeds may cause a slight expansion of the material. Feed also had an impact, with a slight increase in diameter to 6.662 mm at 0.2 mm/rev feed and 12 m/min. Interestingly, at 0.4 mm/rev feeds, the diameter increased significantly, reaching 6.731 mm at 12 m/min and even 6.746 mm at 25 m/min, possibly indicating material deflection or deformation under the higher cutting forces.

In the case of the HDPE 1000 material, at a cutting speed of 12 m/min and a feed rate of 0.1 mm/rev, the hole diameter was the smallest with an average of 6.505 mm, suggesting that the processing parameters are suitable to avoid material expansion. Increasing the cutting speed to 25 m/min resulted in an increase in the average diameter to 6.550 mm, and at 37 m/min, it reached 6.584 mm, indicating that higher cutting speeds may cause a slight expansion of the HDPE 1000 material. Under increased feed conditions, the hole diameter varied less predictably, suggesting that the complex interaction between material and machining conditions may require the fine tuning of parameters (Figure 7).

Concerning the hole diameter variability for PA6, at a cutting speed of 12 m/min and a feed rate of 0.1 mm/rev, the average diameter recorded was 6.604 mm, indicating that lower cutting speeds favor maintaining a diameter close to the desired one. An increase in cutting speed to 25 m/min resulted in a slight expansion of the diameter to an average of 6.640 mm, while an even higher cutting speed of 37 m/min increased the diameter to 6.654 mm. At a feed rate of 0.2 mm/rev, a smaller variation in diameter was observed, with mean values stabilizing around 6.630 mm for all cutting speeds, suggesting that a higher feed rate may mitigate the effect of expansion due to increased cutting speed. However, at a feed rate of 0.4 mm/rev, the diameter increased significantly to an average of 6.667 mm at 12 m/min, showing the combined influence of low cutting speed and high feed rate on material deformation (Figure 8).

For SIKA BLOCK M960, SIKA BLOCK M980, and SIKA BLOCK M700, the data reflects how different cutting speeds and feed rates influence both the size and shape of the processed holes.

At a cutting speed of 12 m/min and feed rate of 0.1 mm/rev, SIKA BLOCK M960 showed an average diameter of 6.712 mm, showing stability at low cutting speeds. At 25 m/min, there was a negligible increase in diameter to 6.717 mm, while at 37 m/min the diameter decreased slightly to 6.707 mm. This behavior suggests that the material responds well to cutting speed variations without significant changes in size (Figure 9).

For SIKA BLOCK M980, at 12 m/min and a 0.1 mm/rev feed rate, the average diameter was 6.697 mm, indicating a slight material response to cutting conditions. Increased cutting speeds at 25 m/min resulted in an average diameter of 6.709 mm, and at 37 m/min, the diameter returned to 6.697 mm, showing that the material has some resistance to deformation under varying conditions (Figure 10).

In the case of the SIKA BLOCK M700 material, at 12 m/min and a 0.1 mm/rev feed, an average diameter of 6.694 mm resulted, with slight variation with increasing cutting speed at 25 m/min (6.705 mm) and 37 m/min (6.689 mm), indicating size consistency despite cutting speed changes (Figure 11).

### 4.2. Cylindricity of Holes

Cylindricity was measured to assess the quality of the hole shape after machining. In general, all materials showed improvements in cylindricity at higher cutting speeds and lower feeds (Figure 12, Figure 13, Figure 14, Figure 15, Figure 16 and Figure 17).

In terms of cylindricity, following POM-C machining, at lower cutting speed and feed settings, POM-C showed a high cylindricity of 0.0606 mm at 12 m/min and a feed of 0.1 mm/rev, due to a combination of elastic and thermal deformation. Improved cylindricity was observed at 25 m/min with a value of 0.0149 mm, suggestive of improved hole quality under these conditions. However, cylindricity decreased at 37 m/min to 0.0175 mm, indicating a stabilization of the material at high cutting speeds. At higher feeds, the trend of cylindricity improvement was reversed, especially at 37 m/min and 0.3 mm/rev, where a significantly higher cylindricity of 0.1165 mm was recorded. At a feed rate of 0.4 mm/rev, there were worrying values of cylindricity, especially at 25 m/min, where it was 0.1013 mm, indicating that increased cutting forces can deteriorate the hole quality significantly.

These results highlight the importance of careful selection of cutting parameters for optimizing hole accuracy and quality in POM-C, a popular material in advanced engineering applications (Figure 12).

As for the hole cylindricity when machining for HDPE 1000, it varied significantly with changing cutting speed and feed rate. At a 12 m/min cutting speed and a 0.1 mm/rev feed, an average cylindricity of 0.0623 mm was observed, which could be improved with optimized machining parameters. An increase in cutting speed to 25 m/min resulted in a lower cylindricity of 0.0840 mm, and at 37 m/min, an average cylindricity of 0.0745 mm was recorded. Interestingly, at a feed rate of 0.3 mm/rev, the cylindricity improved to an average value of 0.0384 mm at a cutting speed of 25 m/min, suggesting that this could be an optimal setting to ensure hole shape accuracy in HDPE 1000. However, at a higher feed rate of 0.4 mm/rev, the cylindricity quality decreased significantly, with an average value of 0.0910 mm at 12 m/min, highlighting the need to maintain moderate feeds to achieve the best results in terms of cylindrical shape (Figure 13).

The cylindricity of the holes machined in PA6 varied in an interesting way depending on the cutting conditions. At a low cutting speed and low feed rate, the average cylindricity was quite good at 0.0462 mm at 12 m/min. Increasing the cutting speed to 25 m/min affected the cylindricity, which increased to 0.0547 mm, but a further increase in cutting speed to 37 m/min reduced the cylindricity to 0.0441 mm, possibly due to a thermal effect counteracting the mechanical deformation. Higher feeds of 0.3 mm/rev reduced cylindricity to 0.0176 mm at 12 m/min, indicating that higher feeds may favor better cylindricity at lower cutting speeds. However, at 37 m/min, cylindricity increased significantly to 0.0418 mm, suggesting that, at high cutting speeds, the effect of advance becomes less predictable. A similar trend was observed at a 0.4 mm/rev feed, where the best cylindricity was at 12 m/min with 0.0491 mm but deteriorated at 37 m/min with 0.1738 mm, indicating that extreme machining conditions can severely compromise the cylindrical shape of holes (Figure 14).

Regarding the cylindricity for the SIKA BLOCK M960 material, at 12 m/min and a feed rate of 0.1 mm/rev, an average value of 0.0120 mm was recorded, indicating a particularly good hole shape. Cylindricity improvement continued at 25 m/min with an average value of 0.0159 mm, but at 37 m/min cylindricity remained stable at 0.0123 mm, suggesting that the material retains good hole quality at higher cutting speeds (Figure 15).

The hole cylindricity for the SIKA BLOCK M980 material showed a larger variation, with the best value of 0.0156 mm at 12 m/min and 0.1 mm/rev feed. At 25 m/min, the cylindricity increased to 0.0251 mm, and at 37 m/min, it reached 0.0572 mm, a sign that the higher cutting speed may negatively influence hole uniformity (Figure 16).

The hole cylindricity of SIKA BLOCK M700 processing recorded an average value of 0.0158 mm at 12 m/min and a feed rate of 0.1 mm/rev, suggesting a good hole shape quality. At higher cutting speeds, the cylindricity was maintained at low values of 0.0171 mm and 0.0248 mm at 25 m/min and 37 m/min, respectively, showing that the material can maintain adequate cylindricity even under more aggressive machining conditions (Figure 17).

### 4.3. Interactions between Parameters

Complex analysis of the interactions between cutting parameters, by applying the ANOVA method, revealed notable influences in the behavior of the response variables, specified in this study by hole diameter and cylindricity. ANOVA decoded the experimental matrix and revealed a significant synergy between cutting speed and feed rate, which together affect machining quality more than their individual impact would suggest.

According to the factorial design, response variables were set with a specific target: an average hole diameter of 6.625 mm and a cylindricity of 0.091 mm. These target values reflect the requirements of high accuracy in polymer processing. The high sensitivity of these variables to changes in cutting parameters was demonstrated by the values recorded in the experiments, ranging from 6.505 mm to 6.746 mm for diameter and from 0.008 mm to 0.174 mm for cylindricity.

The applied statistical model, with a high degree of freedom (a d.f. of 18 for the model and 5 for the experimental error), illustrated an explained variability (R-squared) of 95.97% for the hole diameter, suggesting excellent predictability and control under the chosen parameters. This is supported by the *p*-value of 0.0230, confirming the significance of the model at the 95% confidence level. The interactions between cutting speed and feed rate, as well as the impact of material type (C-factor), were particularly revealing, providing valuable insights into how to optimize machining processes.

Although cylindricity did not show a statistically significant pattern (an R-squared of 79.46% and a *p*-value of 0.5155 for the full model), this does not undermine the importance of the effects analysis. The absence of statistical significance for cylindricity indicates a need for further research to detect subtle interactions between parameters that may influence this critical characteristic.

Finally, extrapolation of the model responses provided a detailed view of the optimal parameter settings, with an index of 93.98%, thus emphasizing that accuracy in the choice of cutting speed and feed rate combinations is the key to achieving the highest quality standards in polymer processing. Meticulous adjustments, guided by robust statistical models and interpreted with technological acumen, translate into superior manufacturing processes, paving the way to excellence in precision manufacturing.

### 4.4. Discussions

At higher feed rates, increased cutting forces can cause mechanical deformation due to material resistance to rapid drill penetration. These additional forces can cause deflection of the drill and compression of the material around the hole, resulting in irregular hole geometry and reduced cylindricity.

In the case of the HDPE 1000 material, higher cutting speeds resulted in an increase in hole diameter, a phenomenon attributed to the elevated temperature sensitivity of HDPE. The material has low thermal conductivity, which means that the heat accumulated during cutting dissipates slowly, causing localized thermal expansion and an increase in hole diameter.

Hole cylindricity was significantly influenced by cutting speed and feed rate, due to the different behavior of the material under the action of these parameters. At lower cutting speeds and higher feed rates, the material may undergo plastic and elastic deformation, leading to poorer cylindricity. In contrast, higher cutting speeds can induce thermal expansion and feed resistance, but, if well balanced, they can improve cylindricity by reducing mechanical deformation.

For polyurethane biresin materials, diameter and cylindricity variations were less pronounced at moderate cutting speeds and feed rates, as these materials have good dimensional stability. Polyurethane biresins have a lower tendency to deform under mechanical and thermal stress, thus maintaining a more accurate hole geometry.

In the case of PA6, higher cutting speeds reduced the hole cylindricity, this effect being due to the high thermal resistance and viscoelastic behavior of the material. PA6 tends to soften and deform at higher temperatures, and at high cutting speeds this viscoelastic behavior can lead to poorer hole uniformity.

The relationship between the mechanical properties of the investigated polymers and the accuracy of the drilled holes is complex and influenced by several factors. Mechanical properties such as tensile strength, modulus of elasticity, and hardness have a significant impact on the behavior of materials during processing.

POM-C: It has high strength and stiffness, which helps maintain precise hole geometry even at variable cutting speeds and feed rates. However, thermal expansion can slightly affect hole diameter.

HDPE 1000: Characterized by high impact strength and low toughness, HDPE 1000 tends to deform more easily under increased cutting forces, thus affecting hole accuracy.

PA6: Its wear resistance and mechanical durability make PA6 sensitive to thermal and mechanical deformation, which can adversely affect hole cylindricity at high cutting speeds.

Polyurethan biresins (SIKA BLOCK M960, M980, and M700): The dimensional stability of these materials contributes to maintaining high hole accuracy, even under aggressive machining conditions. High-density polyurethane (M960, M980) shows better dimensional stability compared to medium-density polyurethane (M700).

By understanding these relationships, the machining process can be optimized to achieve holes with superior accuracy and cylindricity.

## 5. Conclusions

The detailed study of the interactions between cutting parameters and their impact on the quality characteristics of polymer processing has led to some important findings. Through response modeling and ANOVA statistical analysis, the research clearly revealed the significant influence of cutting speed and feed rate on hole diameter and cylindricity.

The results indicated that the optimal cutting parameters vary depending on the type of polymer. For example:-POM-C: Optimal results were obtained at a cutting speed of 25 m/min and a feed rate of 0.2 mm/rev, resulting in minimal deviations in hole diameter and cylindricity;-HDPE 1000: The best performance was observed at a cutting speed of 25 m/min and a feed rate of 0.3 mm/rev, providing a balance between accuracy and surface finish;-PA6: The optimum hole quality was achieved at a cutting speed of 37 m/min and a feed rate of 0.2 mm/rev, minimizing the effects of thermal expansion;-SIKA BLOCK M960 and M980 (high-density polyurethan biresins): The best results were observed at a cutting speed of 12 m/min and an advance rate of 0.1 mm/rev, maintaining high dimensional stability;-SIKA BLOCK M700 (medium-density polyurethane biresin): The optimum cutting parameters were a cutting speed of 25 m/min and a feed rate of 0.2 mm/rev.

Detailed analysis of the influence of cutting speed and feed rate on hole accuracy and cylindricity:-Thermal expansion and mechanical deformation were identified as the main factors affecting these properties.

Establishing the relationship between polymer mechanical properties and machining performance:-Properties such as tensile strength, modulus of elasticity, and hardness significantly influence hole accuracy and quality.

Recommendations for optimizing polymer processing:-Precise adjustment of cutting speed and feed rate are necessary for diverse types of polymers to achieve optimal results in terms of hole diameter and cylindricity.

These findings underline the importance of selecting the appropriate cutting parameters depending on the specific polymer material to achieve high accuracy and minimal deviations in cylindricity.

In addition, although the application of a robust predictive model provided valuable insights, it is important to note that conclusions should be based on the empirical data obtained rather than predictive capabilities. This approach improves the practical applicability of the results to manufacturing processes.

In conclusion, the study demonstrated the combined power of rigorous experimentation and advanced statistical analysis in improving polymer processing. By applying these methodologies, the polymer processing industry can achieve improved quality standards, increased efficiency, and resource optimization. This research provides a solid foundation for future scientific investigations and technological innovations in plastics processing.

## Figures and Tables

**Figure 1 polymers-16-01490-f001:**
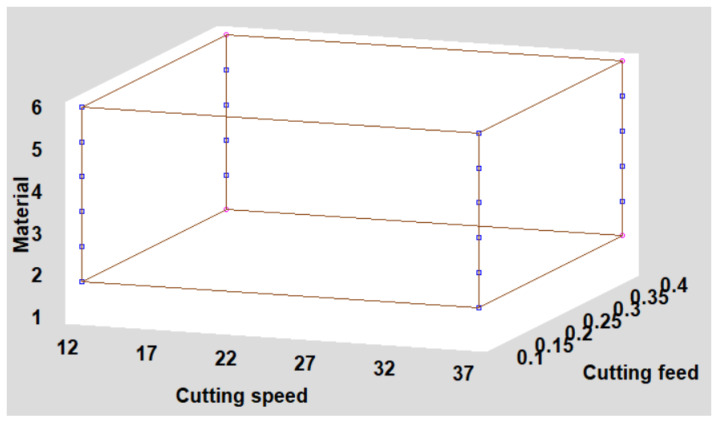
The multilevel factorial design points.

**Figure 2 polymers-16-01490-f002:**
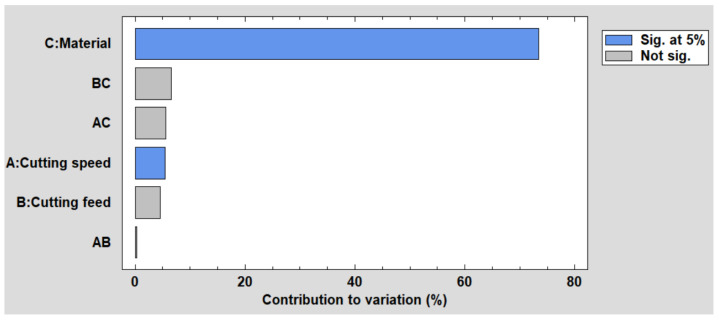
Pareto chart for hole diameter.

**Figure 3 polymers-16-01490-f003:**
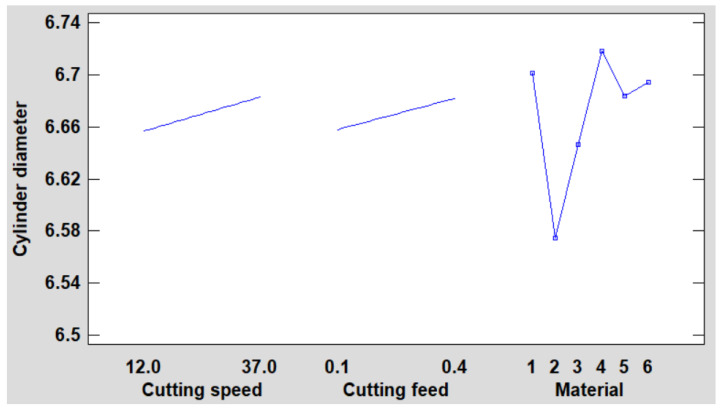
Main effects plot for hole diameter.

**Figure 4 polymers-16-01490-f004:**
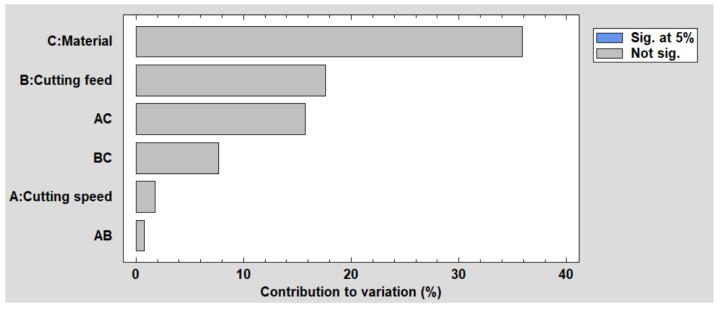
Pareto chart for Cylindricity.

**Figure 5 polymers-16-01490-f005:**
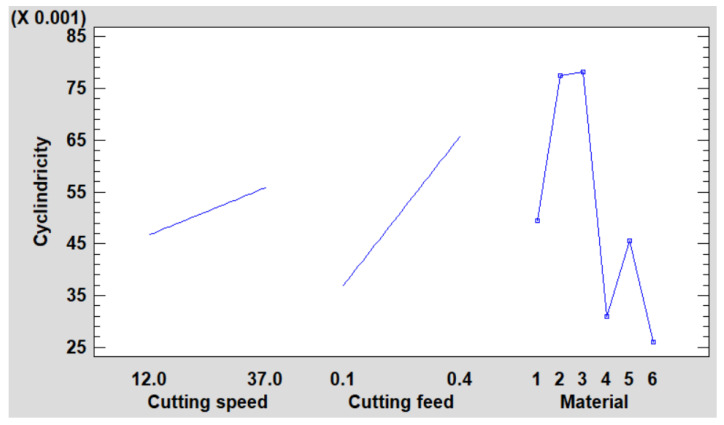
Main effects plot for cylindricity.

**Figure 6 polymers-16-01490-f006:**
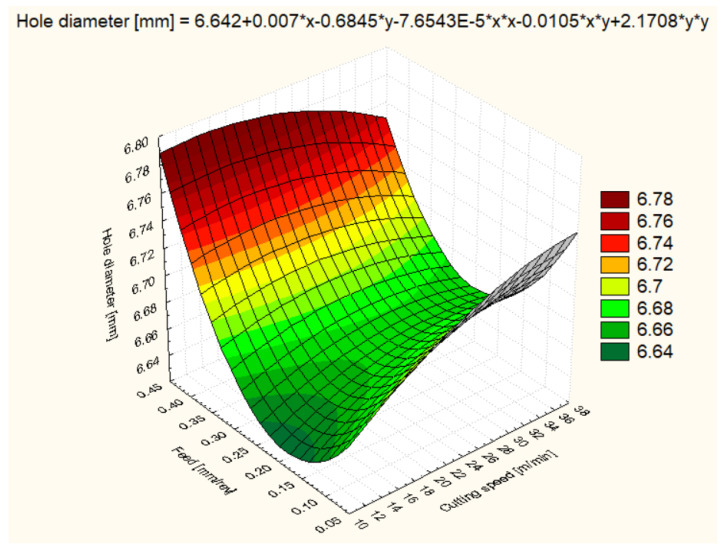
POM-C 3D Surface Plot—Hole Diameter.

**Figure 7 polymers-16-01490-f007:**
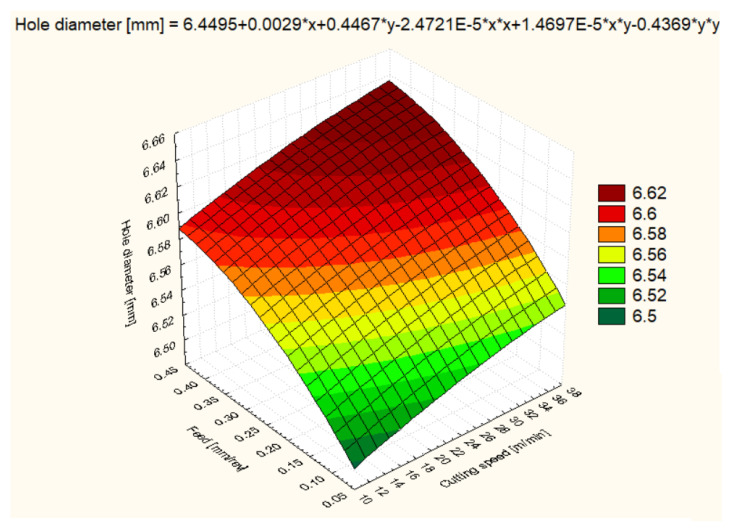
PEHD 1000 3D Surface Plot—Hole Diameter.

**Figure 8 polymers-16-01490-f008:**
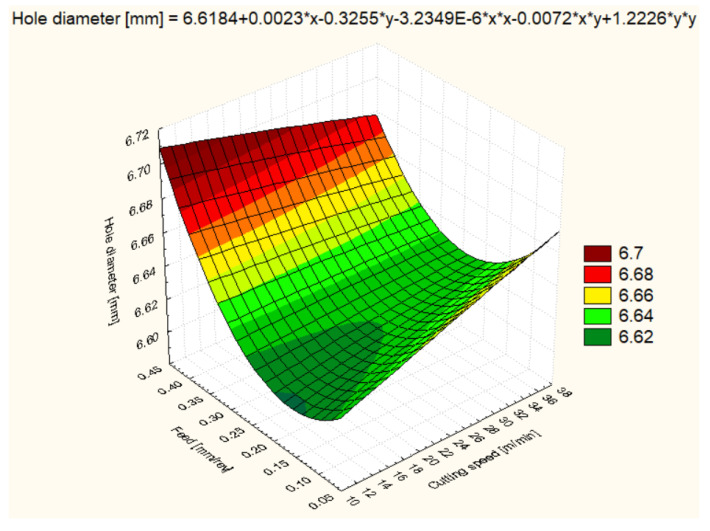
PA6 3D Surface Plot—Hole Diameter.

**Figure 9 polymers-16-01490-f009:**
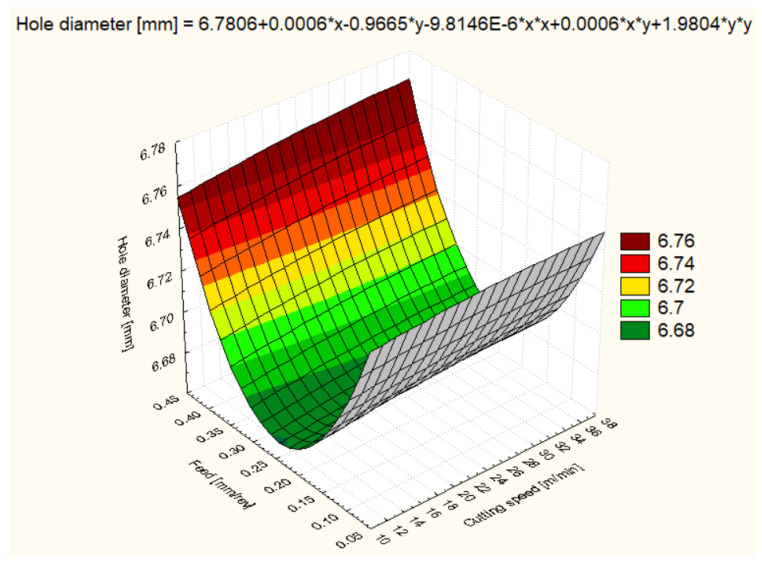
SIKA BLOCK M960 3D Surface Plot—Hole Diameter.

**Figure 10 polymers-16-01490-f010:**
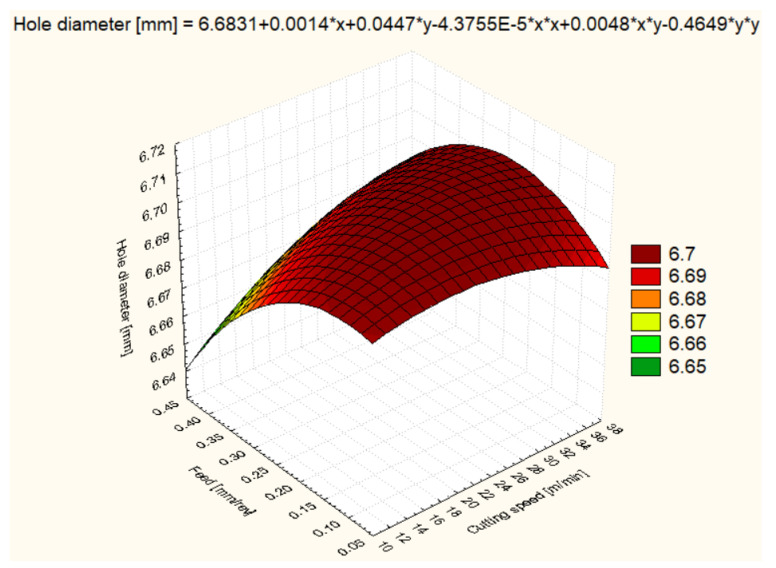
SIKA BLOCK M980 3D Surface Plot—Hole Diameter.

**Figure 11 polymers-16-01490-f011:**
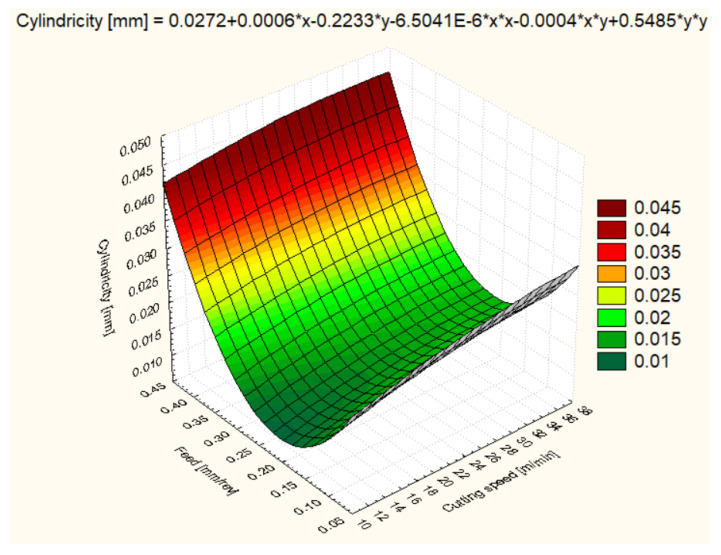
SIKA BLOCK M700 3D Surface Plot—Hole Diameter.

**Figure 12 polymers-16-01490-f012:**
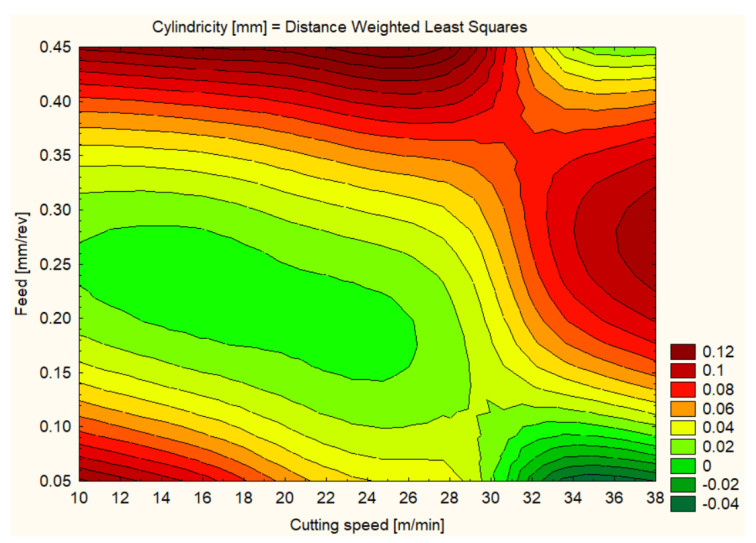
POM-C 3D Contour plot—Cylindricity of Holes.

**Figure 13 polymers-16-01490-f013:**
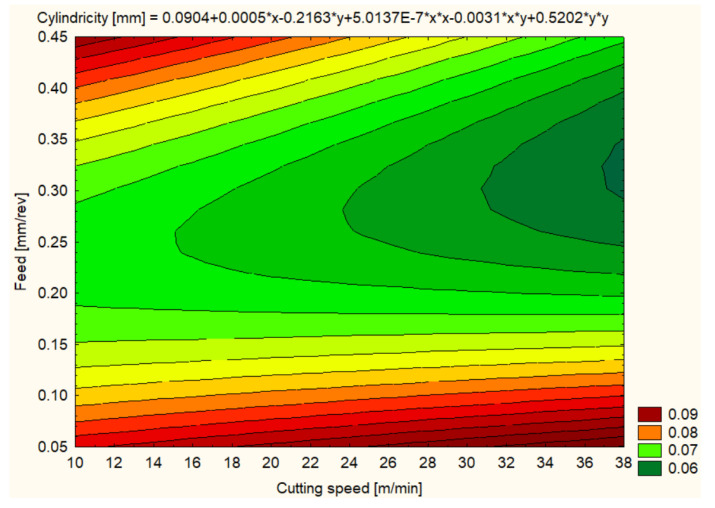
PEHD 1000 3D Contour Plot—Cylindricity of Holes.

**Figure 14 polymers-16-01490-f014:**
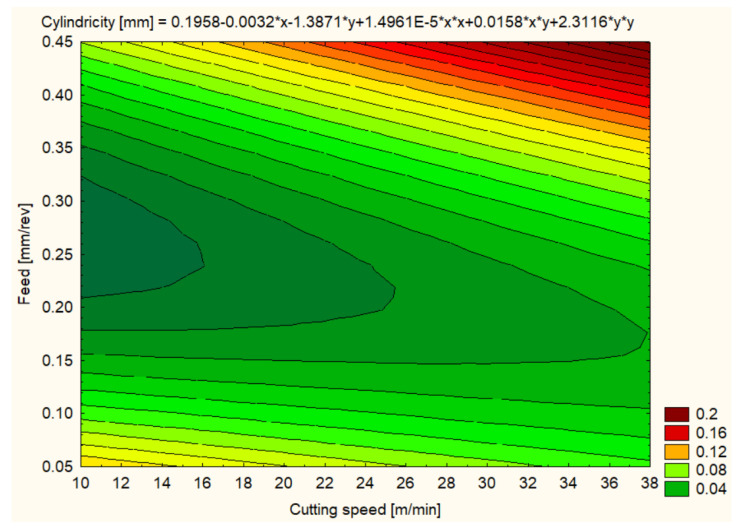
PA6 3D Contour Plot—Cylindricity of Holes.

**Figure 15 polymers-16-01490-f015:**
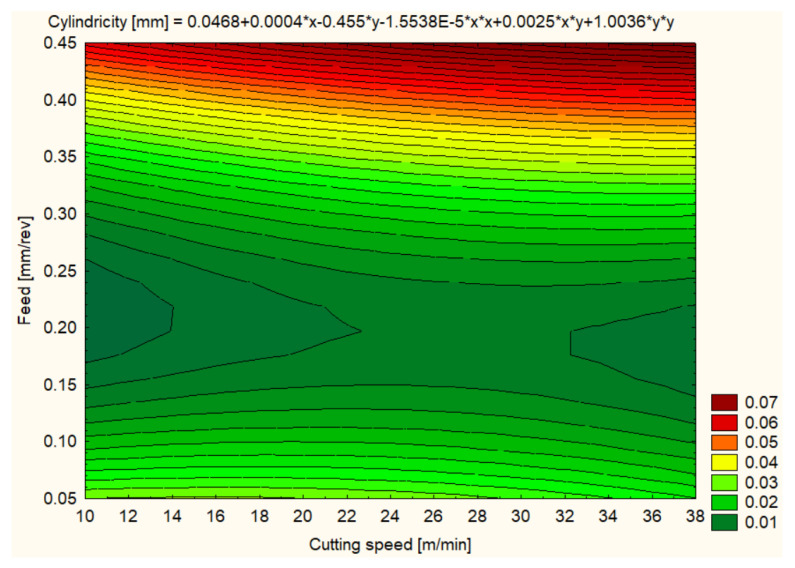
SIKA BLOCK M960 3D Contour Plot—Cylindricity of Holes.

**Figure 16 polymers-16-01490-f016:**
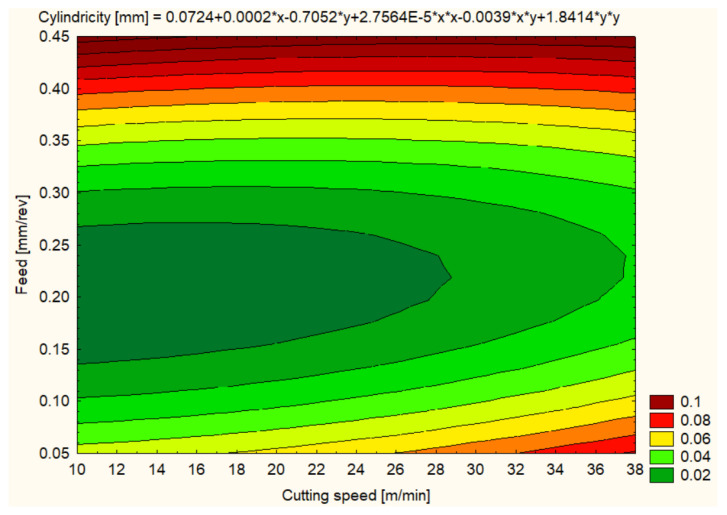
SIKA BLOCK M980 3D Contour Plot—Cylindricity of Holes.

**Figure 17 polymers-16-01490-f017:**
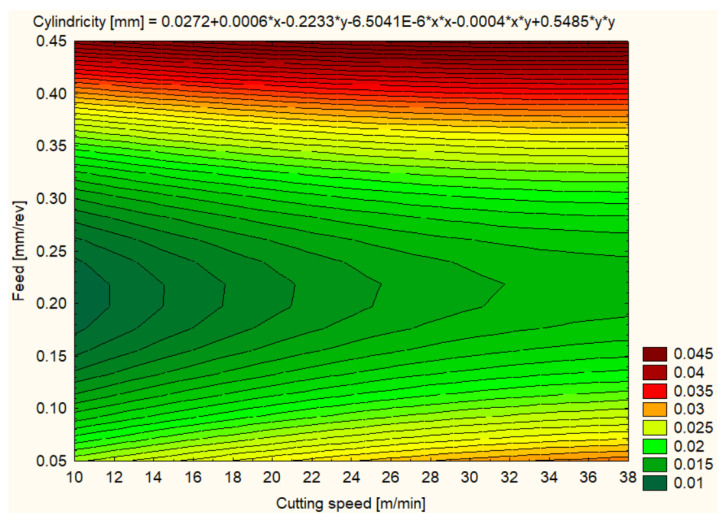
SIKA BLOCK M700 3D Contour Plot—Cylindricity of Holes.

**Table 1 polymers-16-01490-t001:** The mechanical properties of the materials studied.

Material	Tensile Strength (MPa)	Modulus of Elasticity (GPa)	Density (g/cm^3^)	Hardness (Shore D)
POM-C	70	3.5	1.41	85
HDPE 1000	31	1.1	0.95	65
PA6	80	2.8	1.14	80
SIKA BLOCK M960	50	2.0	0.65	70
SIKA BLOCK M980	60	2.2	0.68	75
SIKA BLOCK M700	45	1.8	0.60	60

**Table 2 polymers-16-01490-t002:** Hole diameter and cylindricity measurements.

Material	Cutting Speed [m/min]	Feed [mm/rot]	Hole Diameter [mm]	Cylindricity [mm]
POM-C (Polyacetal-Copolymer, EN ISO 1043)	12	0.1	6.641	0.0606
	25	0.1	6.695	0.0149
	37	0.1	6.712	0.0175
	12	0.2	6.662	0.0104
	25	0.2	6.671	0.0193
	37	0.2	6.671	0.0760
	12	0.3	6.664	0.0142
	25	0.3	6.665	0.0183
	37	0.3	6.653	0.1165
	12	0.4	6.731	0.0683
	25	0.4	6.746	0.1013
	37	0.4	6.721	0.0505
PEHD 1000 (High-Density Polyethylene)	12	0.1	6.505	0.0623
	25	0.1	6.550	0.0840
	37	0.1	6.584	0.0745
	12	0.2	6.565	0.0779
	25	0.2	6.562	0.1015
	37	0.2	6.568	0.0691
	12	0.3	6.601	0.0578
	25	0.3	6.613	0.0384
	37	0.3	6.607	0.0395
	12	0.4	6.566	0.0910
	25	0.4	6.615	0.0525
	37	0.4	6.644	0.0823
PA6 (Polyamide, EN ISO 1874-1)	12	0.1	6.604	0.0462
	25	0.1	6.640	0.0547
	37	0.1	6.654	0.0441
	12	0.2	6.632	0.0229
	25	0.2	6.609	0.0424
	37	0.2	6.631	0.0277
	12	0.3	6.633	0.0176
	25	0.3	6.629	0.0231
	37	0.3	6.620	0.0418
	12	0.4	6.667	0.0491
	25	0.4	6.675	0.0850
	37	0.4	6.661	0.1738
SIKA BLOCK M960 (Polyurethane—High-Density)	12	0.1	6.712	0.0120
	25	0.1	6.717	0.0159
	37	0.1	6.707	0.0123
	12	0.2	6.670	0.0119
	25	0.2	6.686	0.0131
	37	0.2	6.693	0.0115
	12	0.3	6.675	0.0082
	25	0.3	6.675	0.0139
	37	0.3	6.681	0.0107
	12	0.4	6.724	0.0396
	25	0.4	6.726	0.0502
	37	0.4	6.731	0.0597
SIKA BLOCK M980 (Polyurethane—High-Density)	12	0.1	6.697	0.0156
	25	0.1	6.709	0.0251
	37	0.1	6.697	0.0572
	12	0.2	6.693	0.0153
	25	0.2	6.693	0.0194
	37	0.2	6.702	0.0178
	12	0.3	6.705	0.0111
	25	0.3	6.693	0.0179
	37	0.3	6.701	0.0150
	12	0.4	6.644	0.0731
	25	0.4	6.697	0.0648
	37	0.4	6.688	0.0815
SIKA BLOCK M700 (Polyurethane—Medium-Density)	12	0.1	6.694	0.0158
	25	0.1	6.705	0.0171
	37	0.1	6.689	0.0248
	12	0.2	6.685	0.0129
	25	0.2	6.682	0.0142
	37	0.2	6.673	0.0191
	12	0.3	6.685	0.0155
	25	0.3	6.687	0.0185
	37	0.3	6.683	0.0115
	12	0.4	6.697	0.0273
	25	0.4	6.691	0.0363
	37	0.4	6.698	0.0363

**Table 3 polymers-16-01490-t003:** The response variables.

Name	Units	Analyze	Goal	Target	Impact	Sensitivity	Low	High
Hole diameter	mm	Mean	Hit target	6.625	4.0	High	6.505	6.746
Cylindricity	mm	Mean	Hit target	0.091	4.0	High	0.008	0.174

**Table 4 polymers-16-01490-t004:** Experimental factors.

Name	Units	Type	Role	Low	High	Levels
A: Cutting speed	rev/min	Continuous	Controllable	12.0	37.0	
B: Cutting feed	mm/rev	Continuous	Controllable	0.1	0.4	
C: Material		Categorical	Controllable			1, 2, 3, 4, 5, 6

**Table 5 polymers-16-01490-t005:** The experimental design.

Type of	Design	Center Point	Center Point	Design Is	Number of	Total	Total	Error
Factors	Type	Per Block	Placement	Randomized	Replicates	Runs	Blocks	D.F.
Process	Factorial	0	Random	Yes	0	24	1	5

**Table 6 polymers-16-01490-t006:** The experimental results.

Model	Hole Diameter	Cylindricity
Transformation	none	none
Model d.f.	18	18
*p*-value	0.0230	0.5155
Error d.f.	5	5
Stnd. Error	0.0245642	0.0338266
R-squared	95.97	79.46
Adj. R-squared	81.47	5.53

**Table 7 polymers-16-01490-t007:** Extrapolated Response Values.

Step	Desirability	Hole Diameter	Cylindricity
0	0.795956	6.6415	0.0905417
1	0.808353	6.64008	0.0902811
2	0.820757	6.63867	0.0900232
3	0.833169	6.63725	0.089768
4	0.84559	6.63583	0.0895154
5	0.858019	6.63441	0.0892654
6	0.870458	6.63299	0.089018
7	0.882906	6.63157	0.088773
8	0.895364	6.63015	0.0885306
9	0.907834	6.62873	0.0882906
10	0.920314	6.6273	0.088053
11	0.932807	6.62588	0.0878178
12	0.939764	6.62521	0.0877721

**Table 8 polymers-16-01490-t008:** Factor Settings for Extrapolation.

Step	Cutting Speed	Cutting Feed	Material
0	37.0	0.4	2
1	36.625	0.398251	2
2	36.25	0.396509	2
3	35.875	0.394774	2
4	35.5	0.393047	2
5	35.125	0.391325	2
6	34.75	0.389609	2
7	34.375	0.387899	2
8	34.0	0.386195	2
9	33.625	0.384495	2
10	33.25	0.3828	2
11	32.875	0.381109	2
12	32.625	0.381484	2

**Table 9 polymers-16-01490-t009:** ANOVA analysis.

Source	D.F.
Model	18
Total error	5
Lack-of-fit	5
Pure error	0
Total (corr.)	23

**Table 10 polymers-16-01490-t010:** Analysis of Variance for Hole Diameter (mm).

Source	Sum of Squares	D.f.	Mean Square	F-Ratio	*p*-Value
Model	0.0718628	18	0.00399238	6.61647	0.0230
Residual	0.003017	5	0.0006034		
Lack-of-fit		5			
Pure error		0			
Total (corr.)	0.0748798	23			

R-squared = 95.9709%; R-squared (adjusted for d.f.) = 81.466%; standard error of est. = 0.0245642; mean absolute error = 0.00958333; Durbin–Watson statistic = 1.66664 (*p* = 0.1000); Lag 1 residual autocorrelation = 0.117439.

**Table 11 polymers-16-01490-t011:** Analysis of Effects.

Source	Sum of Squares	D.f.	Mean Square	F-Ratio	*p*-Value
A	0.00410817	1	0.00410817	6.80836	0.0477
B	0.00340817	1	0.00340817	5.64827	0.0634
C	0.0549993	5	0.0109999	18.2298	0.0032
AB	0.0001815	1	0.0001815	0.300795	0.6070
AC	0.00420233	5	0.000840467	1.39288	0.3625
BC	0.00496333	5	0.000992667	1.64512	0.2991

Categorical factors: C = material; quantitative factors: A = cutting speed (rev/min); B = cutting feed (mm/rev).

**Table 12 polymers-16-01490-t012:** Analysis of Variance for Cylindricity (mm).

Source	Sum of Squares	D.f.	Mean Square	F-Ratio	*p*-Value
Model	0.0221357	18	0.00122976	1.07474	0.5155
Residual	0.00572121	5	0.00114424		
Lack-of-fit		5			
Pure error		0			
Total (corr.)	0.027857	23			

R-squared = 79.4622%; R-squared (adjusted for d.f.) = 5.52609%; standard error of est. = 0.0338266; mean absolute error = 0.011375; Durbin–Watson statistic = 1.517 (*p* = 0.0381); Lag 1 residual autocorrelation = 0.196756.

**Table 13 polymers-16-01490-t013:** Analysis of Effects.

Source	Sum of Squares	D.f.	Mean Square	F-Ratio	*p*-Value
A	0.000495042	1	0.000495042	0.432637	0.5398
B	0.00490204	1	0.00490204	4.2841	0.0933
C	0.0100072	5	0.00200144	1.74914	0.2772
AB	0.000222042	1	0.000222042	0.194051	0.6780
AC	0.00437621	5	0.000875242	0.76491	0.6121
BC	0.00213321	5	0.000426642	0.37286	0.8486

Categorical factors: C = material; quantitative factors: A = cutting speed (rev/min); B = cutting feed (mm/rev).

## Data Availability

Data are contained within the article.

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
