# Peer review of "Influence of Machining Parameters on the Dimensional Accuracy of Drilled Holes in Engineering Plastics"

_polymers, 2024, doi:10.3390/polym16111490_

Round 1

Reviewer 1 Report

Comments and Suggestions for Authors

The manuscript is on the influence of machining parameters (cutting speed and feed rate) on the dimensional accuracy of drilled holes (hole diameter and cylindricity) in engineering plastics. 6 polymers were considered: POM-C, HDPE 1000, PA6, SIKA BLOCK M960, M980 and M700. The research results are interesting and the manuscript is recommended for publication. A few points should be clarified before publishing.

 1. Section 2. Literature Review. This section requires improvement. The studies carried out in [1-42] are indicated, but the results of these studies are not presented.  

2. Subsection 3.2. Sample Preparation. The text needs improvements:

a) lines 133-137. The number of samples and the number of holes in each of them are mentioned twice.

b) line 138. The phrase “Equipment Configuration” is redundant.

c) lines 139-141. It is better to move this part of the text to the beginning of the subsection “3.3. Cutting Parameters".

d) lines 140-141. What does the “φ” sign mean at 6.824 mm?

3. Lines 148-150. This phrase is not meaningful without specifying the accuracy and repeatability of the results obtained using the equipment mentioned. It is also required to indicate the hole diameter and cylindricity measurements method; what scatter of values along the hole occurs? It makes sense to provide photographs of the resulting holes in polymer samples.

 4. Line 154. How many repetitions took place? 

5. Line 156. Characteristics of the ANOVA program are required, and it is also advisable to provide a reference to the source. 

6. Generally, when describing the results obtained, too little attention is paid to the physical reasons for obtaining a particular result. 

7. The “Conclusions” section requires improvement. It is necessary to describe the results obtained more specifically, or at least indicate the optimal processing parameters found for specific polymers. Also reduce advertising of the proposed approach, since no “accurate predictions” are presented in the conclusions.

 8. References. There is only one reference (#36, 2023) to papers published in Polymers journal for the last three years. The relevance of the topic of the manuscript for the journal can be proven, among other things, by references to papers published in the Polymers journal for the last three years (2022-2024).

Reviewer 2 Report

Comments and Suggestions for Authors

1.     Combine the introduction and literature review as a single section.

2.     Do not use the words like “We”, “Our”, etc. in a scientific report.

3.     The novelty of the work shall be highlighted in the last paragraph of the introduction section.

4.     In the last paragraph of the introduction section, what is done, how, and what was found should be presented.

5.     What is the merit, and motivation for studying the specific materials?

6.     You need to provide details regarding the sources of materials listed in “3.1. Material Selection” such as country, Company, product code, etc.

7.     There is nothing about the mechanical properties of studied materials.

8.     Page 3 “For each material, specimens of standardized sizes were prepared and subjected to the drilling process.” What are the standard sizes??

9.     What standard did you follow for doing the tests?

10. How did you check the hole quality? You need to clarify this.

11. For quality control of drilled holes in composites and polymers, there is some research that introduces criteria to make a comparison between two drilled holes at the top and bottom surfaces of pieces. Here are some of the research introducing these criteria, “Optimization of drilling parameters in composite sandwich structures (PVC core)”, and “Influence of machining parameters on delamination in drilling of GFRP-armour steel sandwich composites”. I recommend you discuss this in the introduction section and see if it is possible to use them as quality criteria in your work.

12. The discussion section needs to discuss the results and explain the changes and the observations.

13. What was the relationship between the mechanical properties of the investigated polymers and the accuracy of the drilled hole? Discuss.

14. Use bullets to highlight the main achievements of the work in the conclusion section.

Round 2

Reviewer 1 Report

Comments and Suggestions for Authors

Accept in present form.

Reviewer 2 Report

Comments and Suggestions for Authors

The paper is accepted in its current form.